# An efficient training scheme for supermodels

Francine Schevenhoven[1,2] and Frank Selten[1]

[1]Royal Netherlands Meteorological Institute, De Bilt, The Netherlands
[2]Geophysical Institute, University of Bergen, Bergen, Norway

*Correspondence to:* F.J. Schevenhoven (francine.schevenhoven@uib.no)

**Abstract.** Weather and climate models have improved steadily over time as witnessed by objective skill scores, although significant model errors remain. Given these imperfect models, predictions might be improved by combining them dynamically into a so-called "supermodel". In this paper a new training scheme to construct such a supermodel is explored using a technique called Cross Pollination in Time (CPT). In the CPT approach the models exchange states during the prediction. The number of possible predictions grows quickly with time and a strategy to retain only a small number of predictions, called pruning, needs to be developed. The method is explored using low-order dynamical systems and applied to a global atmospheric model. The results indicate that the CPT training is efficient and leads to a supermodel with improved forecast quality as compared to the individual models. Due to its computational efficiency, the technique is suited for application to state-of-the art high-dimensional weather and climate models.

## 1 Introduction

Weather and climate models remain imperfect despite continuous model development. For example at middle to high latitudes, the simulated zonal wind stress maximum averaged over an ensemble of state-of-the-art climate models lies 5 to 10 degrees equatorward of observationally based estimates, which means that on average the mid-latitude winds are too strong in the current models (IPCC 2013).

Improving the models is a large research effort. A demanding aspect is that there are many uncertain parameters and approximations because not all physical processes are explicitly resolved. To model and tune all of these aspects requires a huge computational effort. Even if the optimal solution can be achieved, imperfections remain due to the complexity of the climate system with interactions across scales over many orders of magnitude. In order to improve predictions, it often helps to average across model outcomes as model errors tend to average out. Branicki and Majda (2015) provide some evidence that this multi-model ensemble method (MME) indeed improves predictions under certain conditions. However, it is not straightforward which imperfect models and what weights should be used for the MME forecast. Because of this almost all operational MME predictions are based on equal weights.

In contrast to the standard MME, an alternative approach is to let models exchange information during the simulation which leads to new solutions. If the models complement each other, these solutions potentially stay closer to the observed trajectory than the trajectories of the imperfect models individually. Hence both the short term predictability and the climate statistics will improve. The MME approach only combines trajectories from an ensemble of models after the simulation. This can lead

to improved estimates of for instance the true mean state. It cannot however produce trajectories that remain closer to observed trajectories as combining trajectories of different models leads to smoothing.

A successful approach of combining models is found in van den Berge et al. (2011), where combining models into one large supermodel (SUMO) by prescribing connections between model equations is proposed. The connection coefficients are learned from historical observations. The optimization of the coefficients is achieved by minimizing a cost function. In Wiegerinck et al. (2013) it is noted that the size of these coefficients is typically very large. If the connection coefficients are large enough, the system will quickly synchronize into a joint state. This joint state can be described as a weighted superposition of the imperfect models referred to as weighted SUMO.

Since the minimization of a cost function can be computationally very expensive, we propose in this paper a new procedure to construct such a weighted superposition of imperfect models. The weights are learned from observed trajectories. This new learning process is based on an idea proposed by Smith (2001), Cross Pollination in Time (CPT). CPT "crosses" different model trajectories in order to create a larger solution space with trajectories that potentially follow more closely the observed evolution.

Our training method for a weighted supermodel is developed using the Lorenz 63 system (Lorenz, 1963) following the perfect model approach. The model with standard parameter values generates observations and imperfect models are created by perturbing the parameter values. Next, we apply the method to a more chaotic and realistic global atmospheric model with 1449 degrees of freedom by Marshall and Molteni (1993).

Section 2 of this paper explains the training by cross pollination. Applications of the method are described in section 3 for the Lorenz 63 system and in section 4 for the global atmospheric model. The final section discusses the results and provides an outlook to apply the developed approach to state-of-the-art models.

## 2   Training the supermodel

We assume that we have an observed trajectory, called the "truth". The training phase of CPT starts from an observed initial condition in state space. From this initial state, the imperfect models run for one time step each ending in a different state. (See Fig. 1a.) From these endpoints all models run again. Continuing this process leads to a rapid increase of the number of predictions with time. A larger region of the state space thus can be explored. In order to retain only a small number of predictions a pruning step is required. We choose to continue only those predictions that remain closest to the truth, the others are discarded, as is depicted in Fig. 1b.

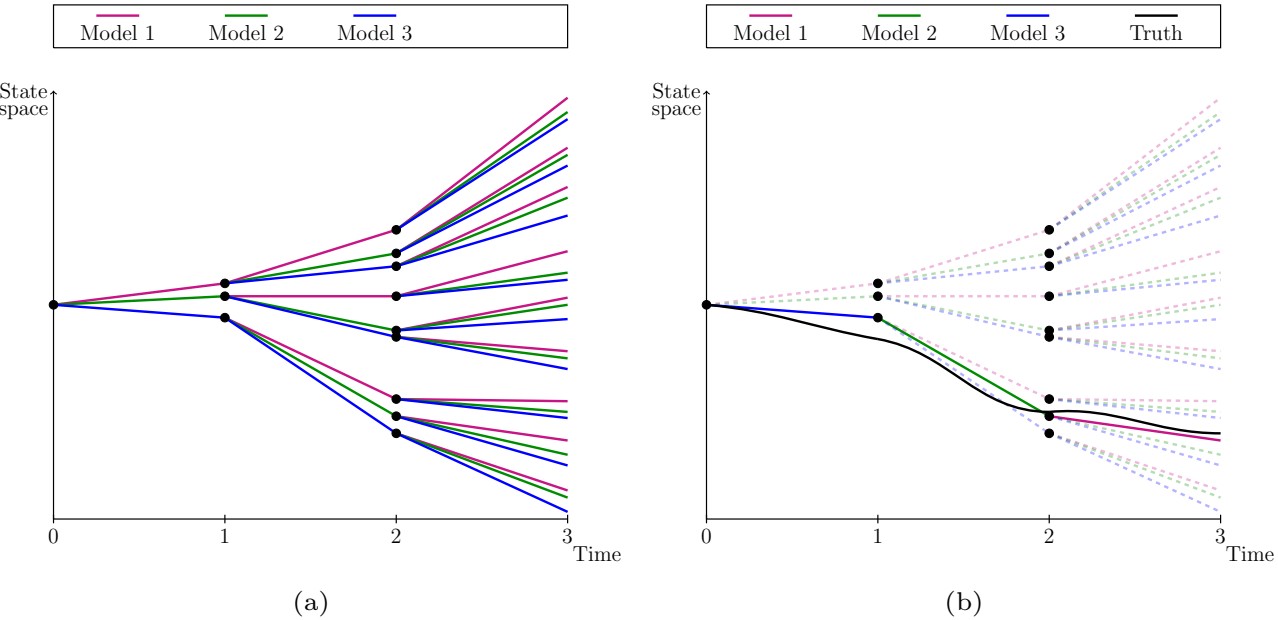

**Figure 1.** Cross Pollination in Time for 3 models, with pruning (right).

## 2.1 Determining weights

In the training phase, for each model, it is counted how often for a particular variable its prediction remains closest to the truth. The probabilities thus obtained can be used as weights for the corresponding time-derivatives of the variables. This superposition of weighted imperfect models forms a supermodel which potentially has improved prediction skill.

## 2.2 Iterative method

In order to obtain convergence towards a supermodel that reflects the truth in the best possible way, the training is carried out iteratively. The first iteration step leads to a first estimate of the weights of the supermodel. In the second iteration step, this supermodel is added as an extra imperfect model. In the subsequent iteration steps the previously obtained supermodel is replaced by the newly obtained supermodel. If the added supermodel is closer to the truth than the initial imperfect models, the constructed trajectory in the CPT procedure receives fewer contributions from the initial imperfect models. Ideally, learning stops when the supermodel remains closer to the truth than the individual imperfect models for all time steps during the training.

## 3 Results Lorenz 63

In the Lorenz 63 system a chaotic attractor appears for certain parameter values. The attractor has the shape of a butterfly and each "butterfly wing" contains an unstable fixed point at its center, around which the trajectories alternately revolve in an unpredictable pattern. The differential equations of the system contain system parameters $\sigma, \rho, \beta$. The state space is described
by coordinates $x, y, z$ (Eq. (1)-(3)).

$$\dot{x} = \sigma(y - x) \tag{1}$$
$$\dot{y} = x(\rho - z) - y \tag{2}$$
$$\dot{z} = xy - \beta z \tag{3}$$

The standard parameter values are $\sigma = 10, \rho = 28$ and $\beta = \frac{8}{3}$. Numerical solutions are obtained by using a fourth-order Runge
Kutta time stepping scheme, with a time step of 0.01.

The observed trajectory is generated by the model with these standard parameter values. Two different imperfect models are created with parameter values that deviate about 30% from the standard parameter values, as denoted in Table 1. In the appendix it is explained why only two different imperfect models are considered and how the imperfect parameter values are chosen.

**Table 1.** Standard and perturbed parameter values for the Lorenz 63 system.

|         | $\sigma$ | $\rho$ | $\beta$       |
|---------|----------|--------|---------------|
| Truth   | 10       | 28     | $\frac{8}{3}$ |
| Model 1 | 12.25    | 19     | 3.3           |
| Model 2 | 7.5      | 35     | 1.9           |

The behavior of these imperfect models is quite different from the truth as can be seen in Fig. 2. Two stable fixed points
characterize the attractor of Model 1. Model 2 has a chaotic attractor that resembles the truth but its mean is shifted towards higher $z$-values.

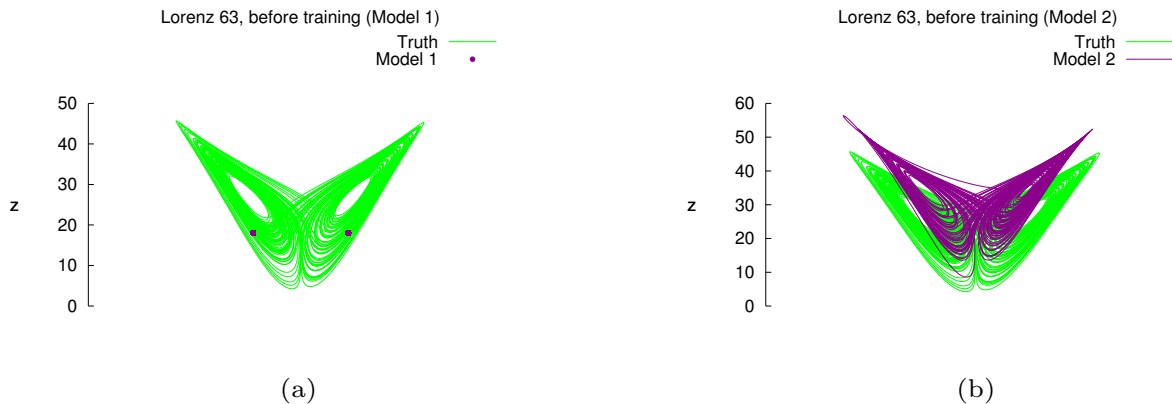

(a)                                           (b)

**Figure 2.** Trajectories of the imperfect models (purple), together with the true trajectory (green).

The training period $T$ is chosen to be 200 time steps, enough to revolve about two times around the unstable fixed points. The number of iterations is 100. The same part of the attractor is used for training in every iteration.

The weights $w_i$, $i \in \{1,2\}$ that are the result of the training phase are listed in Table 2. They determine the superposition of the imperfect models (Eq. (4)-(6)). For all three coordinates $x, y, z$ they sum up to 1.

$$\dot{x}_{super} = \sum_{i=1}^{2} w_i^x \dot{x}_i \tag{4}$$

$$\dot{y}_{super} = \sum_{i=1}^{2} w_i^y \dot{y}_i \tag{5}$$

$$\dot{z}_{super} = \sum_{i=1}^{2} w_i^z \dot{z}_i \tag{6}$$

After 45 iterations, the weights for $\dot{y}$ and $\dot{z}$ do not change anymore. The weights for $\dot{x}$ after 100 iterations are still not constant, but the values differ only from the third decimal onwards.

**Table 2.** Weights of the supermodel of the Lorenz63 system.

| **Model** | $w_i^x$ | $w_i^y$ | $w_i^z$ |
|-----------|---------|---------|---------|
| $i = 1$   | 0.5248  | 0.4385  | 0.5491  |
| $i = 2$   | 0.4752  | 0.5615  | 0.4509  |

10     In the case of the Lorenz 63 system, the superposition of the imperfect Lorenz 63 models forms again a Lorenz 63 system, because the parameter values $\sigma, \rho, \beta$ appear linearly in the differential equations. Hence the supermodel is a Lorenz 63 system

for which the parameter values can be calculated. The supermodel parameters are almost perfect as is shown in Table 3. This is possible because for all three perturbed parameters one of the models has an imperfect parameter value smaller than the standard parameter value and the other model larger (Table 1). Hence, for each of the parameters one can find a linear combination of the imperfect parameter values with positive weights whose sum is equal to one, that represents the standard parameter value (Eq. (7)-(9)).

$$\sigma_{super} = \sum_{i=1}^{2} w_i^x \sigma_i \tag{7}$$

$$\rho_{super} = \sum_{i=1}^{2} w_i^y \rho_i \tag{8}$$

$$\beta_{super} = \sum_{i=1}^{2} w_i^z \beta_i \tag{9}$$

**Table 3.** Parameter values of the truth and the supermodel.

|  | $\sigma$ | $\rho$ | $\beta$ |
|---|---|---|---|
| Truth | 10 | 28 | $\frac{8}{3}$ |
| Supermodel | 9.993 | 27.983 | 2.669 |

If this supermodel is integrated for a long time period, the attractor of the supermodel and the truth look quite similar, as can be seen in Fig. 3.

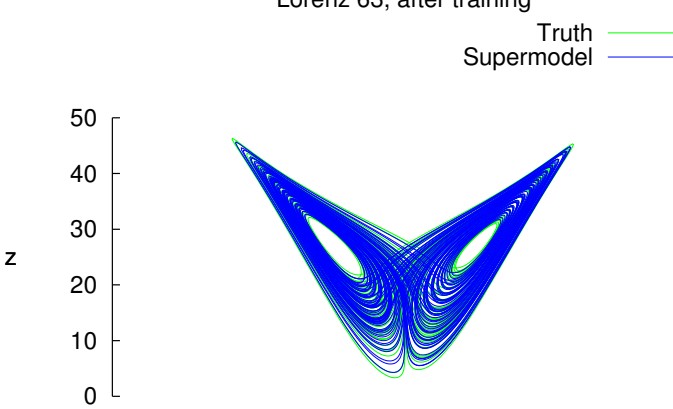

**Figure 3.** Trajectory of the supermodel (blue), together with the true trajectory (green).

## 3.1 Climate measures

Straightforward measures to compare the attractor of the supermodel and the truth are the mean, standard deviation and covariance. The calculation of these statistics is based on 500 runs of 5000 time steps. Also the error estimation of a $95\%$ confidence interval is calculated. In Table 4 can be seen that the statistics of both the true and the supermodel attractor are very similar. Especially the standard deviations for both $x, y$ and $z$ are the same up to the first decimal. The largest differences are in the covariance between $x, z$ and $y, z$. However, these differences are within the $95\%$ uncertainty intervals and are thus not significant. The sizes of all confidence intervals for both the truth and the supermodel are almost identical.

**Table 4.** The mean, standard deviation (SD) and covariance for the truth and the supermodel. Between brackets the $95\%$ error estimation is given.

|  | **Truth** | **Supermodel** |
|---|---|---|
| Mean $x$ | 0.073 (0.099) | 0.033 (0.098) |
| Mean $y$ | 0.073 (0.099) | 0.034 (0.098) |
| Mean $z$ | 23.552 (0.012) | 23.528 (0.012) |
| SD $x$ | 7.843 (0.010) | 7.844 (0.009) |
| SD $y$ | 8.939 (0.011) | 8.942 (0.010) |
| SD $z$ | 8.618 (0.012) | 8.623 (0.012) |
| Cov. $xy$ | 61.529 (0.150) | 61.547 (0.148) |
| Cov. $xz$ | 0.189 (0.266) | 0.057 (0.268) |
| Cov. $yz$ | 0.247 (0.336) | 0.109 (0.334) |

## 3.2 Forecast quality

Along with the measures of the climate statistics of the models, a measure for the quality of the "weather prediction" can also be constructed. This measure reflects the forecast quality of the models on shorter time scales. The squared Euclidean distance between the true trajectory and the trajectory of a model with a slightly perturbed initial condition is calculated and averaged over a number of forecasts, as shown in Fig. 4. On the true attractor, this value converges for large enough forecast time $T$ to a value corresponding to the average distance between two arbitrary states. This distance is used to normalize the measure of the forecast quality.

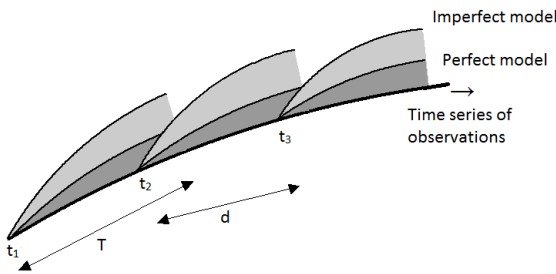

**Figure 4.** Measure of the forecast quality. At times $t_i$ a short integration of time $T$ starts from observed initial conditions and slightly perturbed conditions. The fixed time interval between times $t_i$ is denoted by $d$.

The initial perturbation is chosen in the order of $10^{-1}$. The number of forecasts is equal to 1000 and the distance between the initial states $d$ is 10 time steps. Figure 5 shows that the ability of the supermodel and the true model to predict the observed truth is about the same. In comparison, the imperfect models lose their prediction skill very quickly.

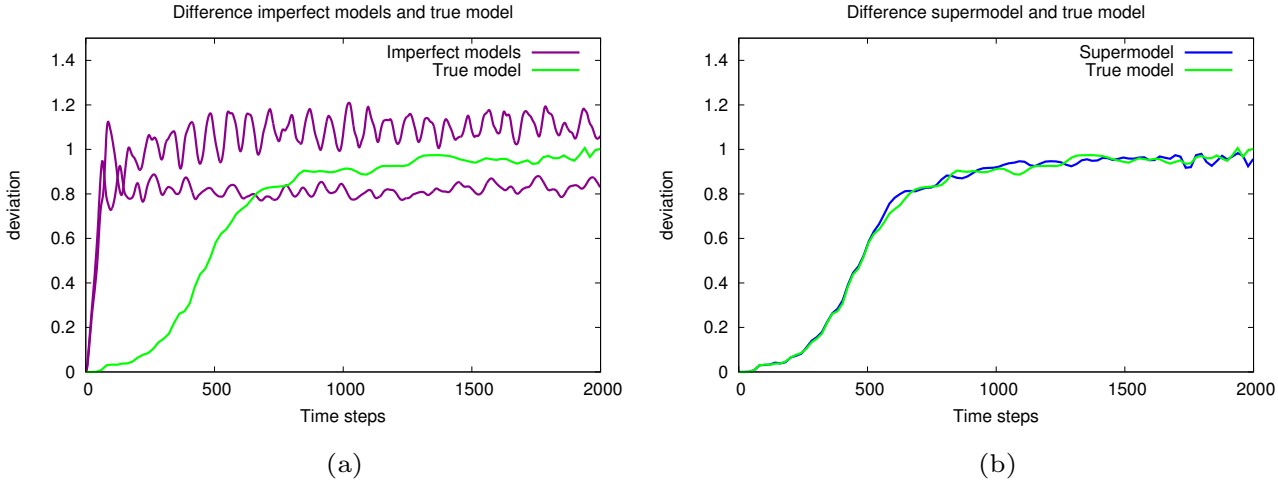

**Figure 5.** Forecast quality of the imperfect Lorenz 63 models (purple) and the supermodel (blue) compared to the true Lorenz 63 model (green).

## 4  Results for a quasi-geostrophic model

Given the encouraging results from the Lorenz 63 system, the CPT method is next applied to a more complex model with 1449 degrees of freedom: a three level quasi-geostrophic global atmosphere model developed by Marshall and Molteni (1993). The model solves the quasi-geostrophic potential vorticity equation on the sphere using a spectral method with spherical harmonic functions. A triangular T21 truncation is used. The performance of this model is quite realistic. According to Corti et al. (1997), the simulation of teleconnections and blockings in the Pacific and Atlantic regions is "surprisingly accurate". The evolution of the quasi geostrophic potential vorticity at the three levels is given by:

$$\dot{q}_1 = \mathcal{J}(\psi_1, q_1) - D_1(\psi_1, \psi_2) + S_1 \tag{10}$$

$$\dot{q}_2 = \mathcal{J}(\psi_2, q_2) - D_2(\psi_1, \psi_2, \psi_3) + S_2 \tag{11}$$

$$\dot{q}_3 = \mathcal{J}(\psi_3, q_3) - D_3(\psi_2, \psi_3) + S_3 \tag{12}$$

where $q$ is the potential vorticity, $\psi$ the streamfunction, $D(\psi)$ a linear operator that represents dissipative terms and $S$ a constant PV source. The index $i$ refers to the pressure level. Here the potential vorticity is defined as:

$$q_1 = \nabla^2 \psi_1 - R_1^{-2}(\psi_1 - \psi_2) + f \tag{13}$$

$$q_2 = \nabla^2 \psi_2 + R_1^{-2}(\psi_1 - \psi_2) - R_2^{-2}(\psi_2 - \psi_3) + f \tag{14}$$

$$q_3 = \nabla^2 \psi_3 + R_2^{-2}(\psi_2 - \psi_3) + f(1 + \frac{h}{H_0}) \tag{15}$$

where $f$ is the Coriolis parameter, $R_1, R_2$ are the Rossby radii of deformation of the 200-500 hPa and 500-800 hPa layer respectively, $h$ is the orographic height and $H_0$ is a scale height. To create different imperfect models, three parameter values are varied:

- $\tau_E$ Time scale in days of the Ekman damping ($\tau_E$ in equation (A11) of Marshall and Molteni (1993) )

- $R_1$ Rossby radius of deformation of the 200-500 hPa layer

- $R_2$ Rossby radius of deformation of the 500-800 hPa layer

Four different imperfect models are used for the CPT training phase, their parameters are denoted in Table 5. The imperfect values of the Rossby radii of deformation are chosen to differ only a few thousandths from the true value since even a small deviation leads to very different behavior of the model. Numerical solutions are obtained by using a fourth-order Runge Kutta time stepping scheme, with a time step of 1/36 day.

**Table 5.** Parameter values of the imperfect QG-models.

|         | $\tau_E$ | $R_1$  | $R_2$  |
|---------|------|--------|--------|
| Truth   | 2.0  | 0.1150 | 0.0720 |
| Model 1 | 1.5  | 0.1165 | 0.0705 |
| Model 2 | 1.5  | 0.1130 | 0.0725 |
| Model 3 | 2.4  | 0.1130 | 0.0705 |
| Model 4 | 2.4  | 0.1165 | 0.0725 |

The training period $T$ is 100 time steps, which corresponds to an integration period of about 3 days. Most of the development of weather systems can be captured within 3 days. The number of iterations is 20. Every iteration a new part of the attractor is used for training by continuation of the observed trajectory, to get a better sampling of the attractor.

In Table 6 the resulting weights for the different levels are shown. After 20 iterations, the weights are not completely
converged, they differ a few percent per iteration, but there is no increasing or decreasing trend. Note that at the 200 hPa level the superposition of models consists of model 1 and model 2 solely. The only parameter with imperfections affecting this level is $R_1$ and the imperfect values of this parameter of model 1 and 4 are the same as well as those of model 2 and 3. Since every time step in the CPT training every model receives the same state, the tendencies of model 1 and 4 are the same at this level and the same holds for model 2 and 3. Therefore the corresponding weights of models 3 and 4 are zero since these are never
chosen during the CPT training.

**Table 6.** Weights of the imperfect QG-models at 200, 500 and 800 hPa.

| **Model** | $w_i^{200}$ | $w_i^{500}$ | $w_i^{800}$ |
|-----------|-------------|-------------|-------------|
| $i = 1$   | 0.653       | 0.217       | 0.093       |
| $i = 2$   | 0.347       | 0.459       | 0.235       |
| $i = 3$   | 0.000       | 0.157       | 0.215       |
| $i = 4$   | 0.000       | 0.167       | 0.457       |

In an additional experiment we left out the imperfect model with the poorest short- and long-term predictability in order to test the hypothesis that the addition of a relatively bad model can still improve the quality of the supermodel solution. The same imperfect models are used. The model with the poorest predictability is model 1 (Table 7), so the supermodel is constructed out of models 2, 3 and 4. Note that these three models still span the same uncertainty range in the three parameters. The same
CPT training phase is applied.

The CPT training provides weights that determine a superposition of models that is capable of following observed trajectories more closely. But to what extend do the values of these weights matter? Is training really necessary? In order to assess this we evaluated the quality of a supermodel with equal weights given to each imperfect model in the superposition.

### 4.1 Climate measures

As measure for the long-term behavior of the quasi-geostrophic model we choose to compare the geostrophic winds of the different models. The potential vorticity calculated by the model determines these winds. The true model, imperfect models and supermodel are integrated over 900 days in a perpetual winter simulation.

As statistical measure (RMSE) the errors in the 900-day average wind strength at the 200 hPa, 500 hPa and 800 hPa level at each location are averaged over the globe:

$$\text{RMSE} = \sqrt{\frac{1}{N}\sum_{i=1}^{N}(\|\boldsymbol{u}_i^{truth}\| - \|\boldsymbol{u}_i^{mod}\|)^2}$$

with $i$ denoting the grid-point, $u$ the zonal wind, $v$ the meridional wind and $N$ the total number of grid-points.

**Table 7.** The root mean squared error of the wind strength (m/s) over 900 winter days. For the true model, the average RMSE is given. Between brackets, the value is given for which $95\%$ of the RMSE values is below that value.

| Model | 800 hPa | 500 hPa | 200 hPa |
|---|---|---|---|
| Model 1 | 1.92 | 1.95 | 2.27 |
| Model 2 | 1.80 | 1.37 | 2.31 |
| Model 3 | 1.10 | 0.90 | 1.79 |
| Model 4 | 1.42 | 1.36 | 2.06 |
| True model | 0.48 (0.65) | 0.78 (0.92) | 1.66 (2.05) |
| Supermodel | 0.45 | 0.80 | 1.77 |
| Supermodel equally weighted | 1.56 | 1.51 | 2.63 |
| Supermodel without worst imperfect model | 1.42 | 1.38 | 2.09 |

We take a Monte Carlo approach to assess the uncertainty of the RMSE values. For 98 different initial conditions a trajectory

of 900 days is computed with the true model. Then the RMSE is calculated for these trajectories with respect to one other true trajectory of observations. The $95\%$ percentile of these values is listed in Table 7. This table reveals that the supermodel with respect to this climate measure is indistinguishable from the true model. The RMSE values of the imperfect models are significantly larger. Note that the supermodel was not trained to reproduce the observed mean state but apparently training on a 3 day time scale is sufficient.

The RMSE values of the supermodel without inclusion of the worst model are comparable with the values of the second best imperfect model, model 4. The RMSE values of the equally weighted supermodel are even worse.

### 4.2 Forecast quality

As was done for the Lorenz 63 system, the forecast quality can be measured by calculating the mean squared error between the true trajectory and the trajectory of a model with a slightly perturbed initial condition and then averaging this over a number

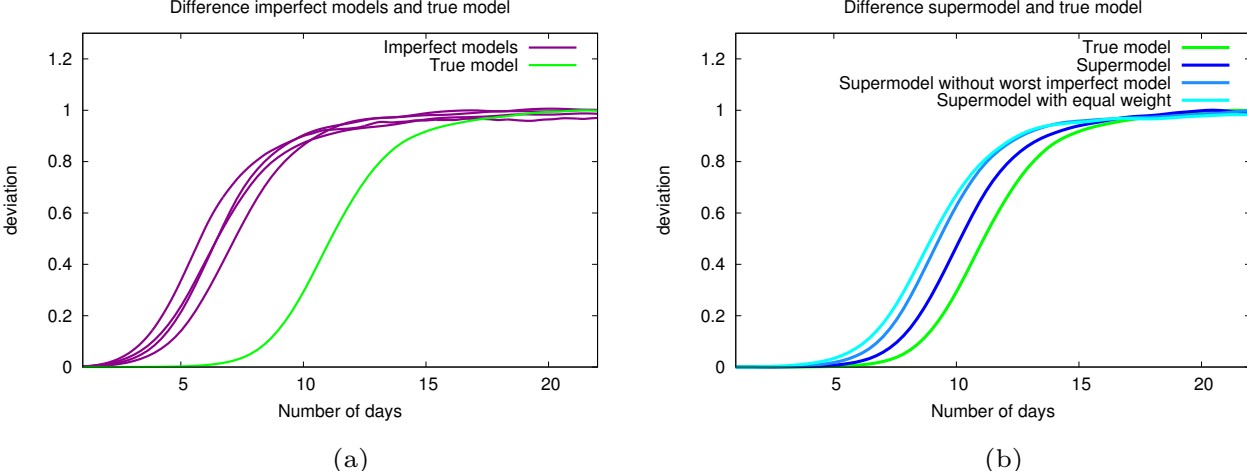

**Figure 6.** Forecast quality of imperfect QG-models (purple), the QG-supermodel (dark blue), the QG-supermodel generated without the worst imperfect model in forecast quality (medium blue) and the QG-supermodel generated with equal weights (light blue) compared to the forecast quality of the true QG-model (green).

of forecasts. The mean squared error is taken over all three levels and all spectral coefficients. The number of forecasts is 100 and the distance $d$ between the initial states is 1000 time steps. In Lorenz (1969) it is mentioned that an initial perturbation in the smallest length scale leads to large errors in all scales after two weeks. For that reason the initial perturbation is chosen in the order of $10^{-4}$ as it leads to an almost complete loss of predictability after 14 days.

The forecast quality of the supermodel is not as good as that of the true model, but the supermodel greatly improves the predictability as compared to the imperfect models (Fig. 6).

The forecast quality of the supermodel without inclusion of the worst model has also improved compared to the imperfect models, but is clearly not as good as the forecast quality of the supermodel with inclusion of this worst model (Fig. 6). Thus inclusion of relatively bad models can still contribute towards a superior supermodel.

The equally weighted supermodel turns out to perform also better than the imperfect models (Fig. 6), but significantly worse than the supermodel with the weights trained by CPT. Hence, training does add value to the quality of the supermodel.

## 5    Conclusions

In this study we have demonstrated that a new training method based on Cross Pollination in Time leads to a dynamical combination of forecast models (a weighted supermodel) with superior forecast quality and improved climatology. The CPT
training is based on short term trajectories only but it turned out that the errors in the climatology are also greatly reduced.

The results indicate that a supermodel with weights trained by CPT can give significantly better predictions than a supermodel consisting of the same imperfect models with equal weights.

State-of-the-art models are far more complex than the examples from this paper but in principle the approach is applicable to state-of-the-art models as well. With an increased number of uncertain parameters, it is to be expected that more imperfect models are required to construct a supermodel with improved prediction skill. This will increase the amount of computation time but if during the CPT training the number of trajectories is pruned back to a single prediction, the computational cost of CPT grows only proportional to the number of imperfect models.

In this study the imperfect models differed in parameter values only but were structurally identical. In reality, imperfect state-of-the-art weather models differ in structure, generally solving different equations on different grids using different numerical methods. In this case, methods from data assimilation might be used in order to cross states between models as is done by Du and Smith (2016). Alternatively, a common state space might be defined with models projecting their states into this common state space and the CPT training limited to this common state space.

In the case when a supermodel solution hardly improves the prediction skill as compared to the imperfect models, one might experiment with the introduction of an additional imperfect model that has in some sense the "opposite" error behavior as compared to the other imperfect models. This additional imperfect model can have worse prediction skill but it might still contribute to a superior supermodel solution. For the quasi-geostrophic atmosphere model in this study it was demonstrated that a model with poor forecast quality still contributed towards an improved supermodel.

A remarkable result of this study is that even if only a relatively small part of the attractor is used for training, the method results in a supermodel with improved climatology. There is evidence in Rodwell and Palmer (2007) that climatological errors develop quickly during the first few days of weather forecasts, implying that a short term training can reduce climatological errors. If this result carries over to the state-of-the-art models then computationally expensive long climate simulations as in Shen et al. (2016) can be avoided during training. Using relatively short integrations only can still improve the climatology of a supermodel.

As indicated above, there are several ways to apply and further develop the CPT training methodology presented in this study. It is not only applicable to weather and climate models, but also to numerical models of other complex systems, for example economical or biological models. Cross Pollination in Time as applied in this paper is a promising approach to combine models dynamically in order to further improve predictions.

## Appendix A

The supermodeling approach only works well if the imperfect models are on "opposite" sides of the truth. We took this into account in the choice of the imperfections. The imperfect values of the parameters and the number of imperfect models is based on the convex hull principle. In one dimension this convex hull principle basically says that if there is one parameter value $\sigma_1$ smaller than the true value $\sigma$ and one parameter value $\sigma_2$ larger than the true value, there are positive weights $w_1, w_2$ such that a linear combination $w_1\sigma_1 + w_2\sigma_2$ is exactly equal to $\sigma$. In case of Lorenz63, the equations for $\dot{x}, \dot{y}$ and $\dot{z}$ each contain only one parameter that appears linearly in the equation. Since we apply different weights for the different equations for $\dot{x}, \dot{y}$ and $\dot{z}$, per equation we need only two imperfect models to be able to reconstruct exactly the true parameter value with positive weights. This convex hull principle can be extended to more dimensions:

**Definition.** *Let $x^1, ..., x^k$ be vectors in $\mathbb{R}^n$ and let $\lambda_1, ..., \lambda_k$ be nonnegative scalars whose sum is unity.*
*(a) The vector $\sum_{i=1}^{k} \lambda_i x^i$ is said to be a **convex combination** of the vectors $x^1, ..., x^k$.*
*(b) The **convex hull** of the vectors $x^1, ..., x^k$ is the set of all convex combinations of these vectors.*

In this definition, the vectors $\mathbf{x}^i, i \in 1, ...k$ represent the imperfect parameter values $x$ per model $i$ and $\lambda_i, i \in 1, ...k$ the corresponding weights. This convex hull generalizes the 'in between' concept for one dimension. To be able to reproduce the $n$-dimensional vector $\mathbf{x}$, it has to lie inside the convex hull of vectors $\mathbf{x}^1, \mathbf{x}^2, ..., \mathbf{x}^k$.

We can write this as a matrix-vector equation, where the last row indicates that the sum of the weights has to be equal to 1 and the vector $\mathbf{x}$ represents the true parameter values:

$$
\begin{pmatrix}
x_1^1 & x_1^2 & \cdots & x_1^{n+1} \\
x_2^1 & x_2^2 & \cdots & x_2^{n+1} \\
\vdots & \vdots & \ddots & \vdots \\
x_n^1 & x_n^2 & \cdots & x_n^{n+1} \\
1 & 1 & \cdots & 1
\end{pmatrix}
\begin{pmatrix}
\lambda_1 \\
\lambda_2 \\
\vdots \\
\lambda_n \\
\lambda_{n+1}
\end{pmatrix}
=
\begin{pmatrix}
x_1 \\
x_2 \\
\vdots \\
x_n \\
1
\end{pmatrix}
$$

For parameter vectors of size $n$, we have $n+1$ constraints, since also the sum of the weights has to equal 1. Hence we know that to be able to reproduce the true parameter vector $\mathbf{x}$, for $n$ parameters that appear linearly in one differential equation for a state variable, $n+1$ linearly independent vectors of these parameters are needed which form a convex hull around the true parameter vector.

For the quasi-geostrophic model, the imperfect parameters do not appear linearly in the equations. Therefore choosing the parameter perturbations such that they form a convex hull around the true parameter values will not necessarily result in a model that reproduces the truth. Nevertheless, in practice we found that this approach still worked well. In this case, choosing the imperfect parameter values on opposite sides of the truth created "opposite" behaviour such that the imperfect models could compensate for each other.

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
