# Peer review of "An efficient training scheme for supermodels"

_Earth System Dynamics, 2017_

## Referee Comment (RC1) · Anonymous Referee #1 · 19 Feb 2017

The authors present a novel approach to multi-model ensemble generation they term "cross pollination in time". The approach is new to me but seems to derive from the earlier work of Smith (2001) and the recent study of Du & Smith (2016). CPT is first illustrated by application to the L63 three component system and then applied to the simple quasi-geostrophic atmospheric model of Marshall & Molteni (1993). The results show that the approach is indeed effective and more encouragingly that inclusion of even a poorly performing model improves the forecast of the supermodel solution. In addition it seems obvious that the approach can be applied generally and in particular to GCMs.

My main criticism is that the method (section 2) is very short and lacking in detail.
Although I can se how to do it, it is not clear exactly how the supermodel weights are calculated or the criteria by which convergence is determined. Given the brevity of the article a bit more technical detail is warranted.

Overall the paper is very well written and clearly articulated. The content is novel and the method promising. Therefore I recommend acceptance after minor revision to more clearly articulate the technical aspects of the approach.

––––––––––––––––––––––––––––––––

---

## Referee Comment (RC2) · G. Hu (Referee) · 21 Feb 2017

This research article proposes a new method to determine the weights of imperfect models in a supermodel, and proves that such a supermodel outperforms every individual imperfect model. Further more the article also shows that supermodel using different weights performs better than using equal weights. The method used to determine weight is called Cross Pollination in Time (CPT). This is a good idea and satisfied results are obtained by applying this method in Lorenz 63 system and a quasi-geostrophic model. The article is clearly structured and the language is fluent and concise, but some parts of the article should be in more detail and clarified. Tables and figures are clear and relevant, although they are not so plentiful. I recommend that this

article needs minor revisions.

Major comment: 1. Section 2 Training the supermodel: my understanding is that every time CPT takes place, it is counted which model is closest to the truth. If we have done 100 times CPT, then we have 100 predictions, and we count for each model in these 100 predictions, how often they have closest predictions to the truth. If I understand correctly, there are several questions. First is CPT interval time should be shown (how often does CPT take place?). Second is whether iterative method is necessary. What if increasing the training time steps or training the model from many different initial conditions. I think these two ways also make the results converge. 2. Page 4, Table 1: if the parameter values for the imperfect models are all smaller or greater than standard values, how well will the supermodel perform? 3. Page 10, Table 5 and page 4, Table 1: an important issue is how to set the parameter values of imperfect models and how to decide the number of imperfect models. These problems are not touched in the article. As shown in conclusion (page 13, line 8 and 9) the number of imperfect models is related to the number of uncertain parameters. The uncertain parameters in Lorenz 63 and quasi-geostrophic model are both 3, but imperfect models used to construct supermodel are 2 and 4, respectively. I wonder if the number of imperfect models depends on the degrees of freedom. As shown in Table 5 each two imperfect models have a same parameter value, why? Is it possible to randomly select parameter values for imperfect models with given error. 4. Page 12, 4.3 Use of fewer imperfect models: I am interested in the difference between supermodel without the worst imperfect model and supermodel without the best imperfect model. In such a way, it can be clarified whether the accuracy of imperfect model affects the performance of supermodel. Clearly, the number of imperfect models influences the skill of the supermodel. While the relationship is unclear. The differences between supermodel and truth as a function of number of imperfect models can used to illustrate this relationship.

Minor comment: 1. Abstract, page 1, line 7 and 8: the supermodel obtained by the CPT training should not only compared to the individual imperfect models, it is not the topic

of this article. As indicated by the title: the method improves the supermodel, so the comparison should between the supermodels with different weights. 2. Introduction: I'm wondering if there are still other methods have been used to determine the weights except the example in the second paragraph of page 2. And I also want to know the advantages of CPT, since the multi-model ensemble method (MME) can also enlarge the state space. 3. Page 4, line 12: why choose deviation 30%? 4. Page 5, Figure 2 and page 7, Figure 3: I suggest figure of 3D-phase space. 5. Page 7, 3.1 Climate measures: I suggest using probability density function and autocorrelation function to compare the supermodel and truth, not only the mean, standard deviation and covariance. 6. Page 9, Figure 5: (a) of Figure 5 is not necessary. I suggest combine (a) and (b) of Figure 5. 7. Page 9, line 15 and page 10, line 1: in the experiments with Lorenz 63, the model 1 has two stable fixed points and very different behavior compared to standard Lorenz 63 system. But it still contributes to form supermodel. 8. Page 11, Table 7: I suggest add tables for other height (200hPa, 500hPa) and in table supermodel with same weights should be included. 9. Page 11, line 10: what is the minimal time steps for training? If the training time steps increases, does the weights change? 10. The supermodel with different weights performs better than supermodel with equal weights, but it requires computational time. Increasing the number of imperfect models also improves the skill of the supermodel with equal weights. So we need to compare in which way the computational demand is lower in certain application.

Typos and misspelling: 1. Page 1, end of line 8: "state-of-the art", "-" before "art" is missing 2. Page 1, line 16: add "," before because 3. Page 4, title of section 3: "Results Lorenz 63", "for" missing

---

## Author Comment (AC1) · 31 Mar 2017

We would like to thank the reviewer for his helpful comments. We revised our manuscript and will give a reply to the reviewer's comments below.

*My main criticism is that the method (section 2) is very short and lacking in detail. Although I can see how to do it, it is not clear exactly how the supermodel weights are calculated or the criteria by which convergence is determined. Given the brevity of the article a bit more technical detail is warranted.*

The supermodel weights are calculated just by counting how often a certain imperfect model is chosen during the CPT training. For example, if a model is chosen $30\%$ of

the time, then the corresponding weight will be 0.3. Convergence is obtained when during the CPT training every time step the supermodel is the model that is closest to the truth. This perfect convergence does not happen with our examples. The majority of time steps the supermodel is chosen, but sometimes the other imperfect models are closer to the truth. However, the obtained weights do not significantly change after certain time.

More comments on technical details can be found in the reply to reviewer Guannan Hu.

---

## Author Comment (AC2) · 31 Mar 2017

Reply to reviewer Guannan Hu.

We would like to thank Guannan Hu for his very helpful comments. We revised our manuscript and will give a pointwise reply to Guannan Hu's comments below.

**Major comments**

**1**

*Section 2 Training the supermodel: my understanding is that every time CPT takes place, it is counted which model is closest to the truth. If we have done 100 times CPT, then we have 100 predictions, and we count for each model in these 100 predictions, how often they have closest prediction to the truth. If I understand correctly, there are several questions. First is CPT interval time should be shown (how often does CPT take place?). Second is whether iterative method is necessary. What if increasing the training timesteps or training the model from many different initial conditions. I think these two ways also make the results converge.*

The CPT step takes place every time step.
Increasing the number of training steps will not always give a better estimation for the weights. For example, it can happen that during training the true trajectory cannot be followed anymore by the CPT trajectory, the CPT trajectory can diverge from the true trajectory. In that case, the resulting weights will probably not give a supermodel with an improved forecast quality. For the quasi-geostrophic model we have started the training from many different initial points. In that case we train on a larger part of the attractor. If the supermodel is closer to the truth than the initial imperfect models then adding the supermodel as an extra imperfect model in a next iteration step enables the CPT trajectory to follow the true trajectory more closely and fewer strong adjustments will be needed. Hence, this can result in a better estimation of the weights.

**2**

*Page 4, Table 1 if the parameter values for the imperfect models are all smaller or greater than standard values, how well will the supermodel perform?*

The supermodeling approach only works well if the imperfect models are on "opposite" sides of the truth. We took this into account in the choice of the imperfections. The imperfect values of the parameters and the number of imperfect models is based on the convex hull principle. In one dimension this convex hull principle basically says that if there is one parameter value $\sigma_1$ smaller than the true value $\sigma$ and one parameter value $\sigma_2$ larger than the true value, there are positive weights $w_1, w_2$ such that a linear combination $w_1\sigma_1 + w_2\sigma_2$ can be exactly equal to $\sigma$. In case of Lorenz63, the equations for $\dot{x}, \dot{y}$ and $\dot{z}$ contain each only one parameter that appears linearly in the equation. Since we apply different weights for the different equations for $\dot{x}, \dot{y}$ and $\dot{z}$, per equation we need only two imperfect models to be able to reconstruct exactly the true parameter value with positive weights. This convex hull principle can be extended to more dimensions:

**Definition.** *Let $\boldsymbol{x}^1, ..., \boldsymbol{x}^k$ be vectors in $\mathbb{R}^n$ and let $\lambda_1, ..., \lambda_k$ be nonnegative scalars whose sum is unity.*
*(a) The vector $\sum_{i=1}^{k} \lambda_i \boldsymbol{x}^i$ is said to be a **convex combination** of the vectors $\boldsymbol{x}^1, ..., \boldsymbol{x}^k$.*
*(b) The **convex hull** of the vectors $\boldsymbol{x}^1, ..., \boldsymbol{x}^k$ is the set of all convex combinations of these vectors.*

In this definition, the vectors $\mathbf{x}^i, i \in 1, ...k$ represent the imperfect parameter values $x$ per equation $i$ and $\lambda_i, i \in 1, ...k$ the corresponding weights. This convex hull generalizes the 'in between' concept for one dimension. To be able to reproduce the $n$-dimensional vector $\mathbf{x}$, it has to lie inside the convex hull of vectors $\mathbf{x}^1, \mathbf{x}^2, ..., \mathbf{x}^k$.
We can write this as a matrix-vector equation, where the last row indicates that the sum of the weights has to be equal to 1 and the vector $\mathbf{x}$ represents the true parameter values:

$$\begin{pmatrix} x_1^1 & x_1^2 & \cdots & x_1^{n+1} \\ x_2^1 & x_2^2 & \cdots & x_2^{n+1} \\ \vdots & \vdots & \ddots & \vdots \\ x_n^1 & x_n^2 & \cdots & x_n^{n+1} \\ 1 & 1 & \cdots & 1 \end{pmatrix} \begin{pmatrix} \lambda_1 \\ \lambda_2 \\ \vdots \\ \lambda_n \\ \lambda_{n+1} \end{pmatrix} = \begin{pmatrix} x_1 \\ x_2 \\ \vdots \\ x_n \\ 1 \end{pmatrix}$$

For parameter vectors of size $n$, we have $n + 1$ constraints, since also the sum of the weights has to equal 1. Hence we know that to be able to reproduce the true parameter vector $\mathbf{x}$, for $n$ parameters that appear linearly in one differential equation for a state variable, $n + 1$ linearly independent vectors of these parameters are needed which form a convex hull around the true parameter vector.

For the quasi-geostrophic model, the imperfect parameters do not appear linearly in the equations. Therefore choosing the parameter perturbations such that they form a convex hull around the true parameter values will not necessarily result in model that represents the truth. Nevertheless, in practice we found that this approach worked well. In this case, choosing the imperfect parameter values on opposite sides of the truth created "opposite" behaviour such that the imperfect models could compensate for each other.

**3**

*Page 10, Table 5 and page 4, Table 1: an important issue is how to set the parameter values of imperfect models and how to decide the number of imperfect models. These problems are not touched in the article. As shown in conclusion (page 13, line 8 and 9) the number of imperfect models is related to the number of uncertain parameters. The uncertain parameters in Lorenz 63 and quasi-geostrophic model are both 3, but imperfect models used to construct supermodel are 2 and 4, respectively. I wonder if the number of imperfect modes depends on the degrees of freedom. As shown in Table 5 each two imperfect models have a same parameter value, why? Is it possible to randomly select parameter values for imperfect models with given error.*

In the previous comment we explained how to set the parameter values of the imperfect models and how to decide on the number of imperfect models. As you pointed out, we think it is important to touch these problems in the article, therefore we will add an appendix that explains this.

**4**

*Page 12, 4.3 Use of fewer imperfect models: I am interested in the difference between supermodel without the worst imperfect model and supermodel without the best imperfect model. In such a way, it can be clarified whether the accuracy of imperfect model affects the performance of supermodel. Clearly the number of imperfect models influences the skill of the supermodel. While the relationship is unclear. The differences between supermodel and truth as a function of number of imperfect models can used to illustrate this relationship.*

The number of imperfect models influences the skill of the supermodel. Since the number of imperfect models is not generalizable for every model, because parameters will not always appear linearly in the equations, we did not investigate the differences between the supermodel and truth as a function of number of imperfect models.

**Minor comments**

**1**

*Abstract, page 1, line 7 and 8 the supermodel obtained by the CPT training should not only compared to the individual imperfect models, it is not the topic of this article. As indicated by the title the method improves the supermodel, so the comparison should between the supermodels with different weights.*

The scope of this paper is to show that a supermodel constructed by the CPT training can outperform

individual imperfect models. Also it is shown that the CPT training has effect compared to a supermodel with equal weights. It was not the purpose to compare the method with other methods that give a weighted supermodel. We agree that the title is misleading, hence we changed it into: "An efficient training scheme for supermodels".

**2**

*Introduction: I'm wondering if there are still other methods have been used to determine the weights except the example in the second paragraph of page 2. And I also want to know the advantages of CPT, since the multi-model ensemble method (MME) can also enlarge the state space.*

There are no other examples of methods to determine the weights for a supermodel, to my knowledge. In this study we use CPT to find optimal weights for a weighted supermodel. CPT generates new trajectories in contrast to a Multi Model Ensemble (MME) approach that only combines trajectories from an ensemble of models. CPT accomplishes this by continuing a trajectory from one model using another model and creating hybrid trajectories this way that are trained to lie closer to the truth. After training, the supermodel climate statistics are closer to the truth since trajectories more closely follow true trajectories. In a MME approach, combining model statistics leads to improved estimates of for instance the true mean state. But in the supermodel approach, not only is the mean state of the supermodel closer to the truth but in addition the supermodel trajectories from which this mean state is calculated are closer to observed trajectories. In a MME approach this is not the case. Combining trajectories of different models leads to smoothing since weather systems and fronts are in different locations.

**3**

*Page 4, line 12: why choose deviation 30%?*

The deviation of 30% in the imperfect parameter values is chosen because then the imperfect models have completely different behaviour compared to the true model. For the Lorenz 63 model we showed that even for this large deviation we are still able to construct a supermodel that reflects the truth quite well.

**4**

*Page 5, Figure 2 and page 7, Figure 3: I suggest figure of 3D-phase space.*

Thank you for the suggestion but we think that a figure that shows more of the 3D structure will be less clear than this figure.

**5**

*Page 7, 3.1 Climate measures: I suggest using probability density function and autocorrelation function to compare the supermodel and truth, not only the mean, standard deviation and covariance.*

The supermodel constructed out of the imperfect Lorenz 63 models is again a Lorenz 63 model. Since the supermodel parameters $\rho, \sigma, \beta$ are very close to the true parameter values and since there is no bifurcation for this system close by the true parameter values, it is to be expected that the supermodel has an attractor that is very similar to the true attractor. To illustrate this, we only used mean, variance and covariance, and since there are no significant differences between the two attractors, we did not use any other statistical measures.

**6**

*Page 9, Figure 5: (a) of Figure 5 is not necessary. I suggest combine (a) and (b) of Figure 5.*

In the case of the forecast quality of the Lorenz 63 model,(a) and (b) of Figure 5 could be combined, since there are not that many graphs. However, we used the same figures for the forecast quality of the quasi-geostrophic model. For this model we had more graphs that cannot be merged into one figure, that

would be unclear. Hence we chose to separate the figure for the forecast quality for the imperfect models from the forecast quality for the supermodels for both Lorenz 63 and the quasi-geostrophic model.

**7**

*Page 9, line 15 and page 10, line 1: in the experiments with Lorenz 63, the model 1 has two stable fixed points and very different behavior compared to standard Lorenz 63 system. But it still contributes to form supermodel.*

For the quasi geostrophic model we chose more realistic imperfect models than for the Lorenz 63 system. The Rossby radii of deformation are known quite well. Hence we chose imperfect parameter values that cause quite different behavior compared to the truth (the imperfect models perform much poorer in our short and long term measures than the true model) but this was accomplished by imperfect Rossby radii that are not that far from the true values.

**8**

*Page 11, Table 7: I suggest add tables for other height (200hPa, 500hPa) and in table supermodel with same weights should be included.*

We included the RMSE for a supermodel with equal weights and the RMSE for the supermodel constructed without the imperfect model with the poorest forecast quality and we added tables for 200hPa and 500hPa. The supermodel with the equal weights performs better than imperfect model 1 and 2, but worse than imperfect model 3 and 4 and the supermodel. The supermodel constructed without the imperfect model with the poorest climate and forecast quality performs slightly better than the supermodel with equal weights, but the errors are significant compared to the perfect model. The supermodel is also for the 200hPa and 500hPa level indistinguishable from the perfect model.

**9**

*Page 11, line 1: what is the minimal time steps for training? If the training time steps increases, do the weights change?*

Already after integrating for half a day, the estimation of the weights is not that different from the final weights, they differ around 10%. If the number of training steps is more than 3 days, the weights do not significantly change, they differ a few percent but there is no structural deviation from the obtained weights mentioned in this paper.

**10**

*The supermodel with different weights performs better than supermodel with equal weights, but it requires computational time. Increasing the number of imperfect models also improves the skill of the supermodel with equal weights. So we need to compare in which way the computational demand is slower in certain application.*

Increasing the number of imperfect models can help to improve the quality of supermodel with equal weights, but this does not necessarily have to be the case. We then have to decide which imperfect parameters we will use, which also takes time. Running a supermodel that consist of more imperfect models will also take more computational time. A supermodel with equal weights that consists of the same amount of imperfect models as a supermodel with trained weights costs the same amount of computation time. Training is relatively cheap and only has to be done once.